# Healing of Experimental Periodontal Defects Following Treatment with Fibroblast Growth Factor-2 and Deproteinized Bovine Bone Mineral

**DOI:** 10.3390/biom11060805

**Published:** 2021-05-29

**Authors:** Tasuku Murakami, Daisuke Matsugami, Wataru Yoshida, Kentaro Imamura, Takahiro Bizenjima, Fumi Seshima, Atsushi Saito

**Affiliations:** 1Department of Periodontology, Tokyo Dental College, Chiyoda-ku, Tokyo 1010061, Japan; murakamitasuku@tdc.ac.jp (T.M.); matsugamidaisuke@tdc.ac.jp (D.M.); yoshidawataru@tdc.ac.jp (W.Y.); imamurakentarou@tdc.ac.jp (K.I.); seshimaf@tdc.ac.jp (F.S.); 2Oral Health Science Center, Tokyo Dental College, Chiyoda-ku, Tokyo 1010061, Japan; 3Chiba Dental Center, Tokyo Dental College, Mihama-ku, Chiba 2618502, Japan; bizenjimatakahiro@tdc.ac.jp

**Keywords:** basic fibroblast growth factor (FGF-2), deproteinized bovine bone mineral (DBBM), periodontitis, periodontal regeneration, cell proliferation, angiogenesis, osteoblast differentiation

## Abstract

The aim of this study was to investigate the effects of fibroblast growth factor (FGF)-2 used in combination with deproteinized bovine bone mineral (DBBM) on the healing of experimental periodontal defects. Periodontal defects created in rats were treated by FGF-2, DBBM, FGF-2 + DBBM, or left unfilled. Microcomputed tomography, histological, and immunohistochemical examinations were used to evaluate healing. In vitro cell viability/proliferation on DBBM with/without FGF-2 was assessed by WST-1. Cell behavior was analyzed using scanning electron and confocal laser scanning microscopy. Osteogenic differentiation was evaluated by staining with alkaline phosphatase and alizarin red. Bone volume fraction was significantly greater in FGF-2 and FGF-2 + DBBM groups than in other groups at 2 and 4 weeks postoperatively. In histological assessment, newly formed bone in FGF-2 and FGF-2 + DBBM groups appeared to be greater than other groups. Significantly greater levels of proliferating cell nuclear antigen-, vascular endothelial growth factor-, and osterix-positive cells were observed in FGF-2 and FGF-2 + DBBM groups compared to Unfilled group. In vitro, addition of FGF-2 to DBBM promoted cell viability/proliferation, attachment/spreading, and osteogenic differentiation. The combination therapy using FGF-2 and DBBM was similarly effective as FGF-2 alone in the healing of experimental periodontal defects. In certain bone defect configurations, the combined use of FGF-2 and DBBM may enhance healing via promotion of cell proliferation, angiogenesis, and osteogenic differentiation.

## 1. Introduction

A successful outcome of periodontal tissue engineering requires the following factors: biological agent, scaffold, progenitor cells, and adequate blood supply [1]. One promising biologics, fibroblast growth factor (FGF)-2, induces proliferation and migration of periodontal ligament–derived cells (PDLCs), which play an important role in periodontal regeneration [2,3,4,5]. Based on the favorable results from basic studies and clinical trials [6,7,8,9,10,11], 0.3% recombinant human FGF-2 was formally approved in 2016 and has been used as a periodontal regenerative medicine in Japan [12].

Deproteinized bovine bone mineral (DBBM) possesses osteoconductive properties and functions as a scaffold for bone regeneration [13,14,15]. New bone formation following the use of enamel matrix derivative (EMD) combined with DBBM was greater than DBBM alone in pre-clinical studies [16,17,18]. In vivo studies showed that use of FGF-2 with beta-tricalcium phosphate (β-TCP) promotes periodontal healing [19,20]. It was thus hypothesized that the use of FGF-2 with bone substitutes such as DBBM could yield enhanced regenerative outcome. To test this, we conducted a randomized controlled trial comparing the use of FGF-2 + DBBM to FGF-2 alone in the treatment of intrabony periodontal defects [12,21]. Although adding DBBM to FGF-2 yielded greater radiographic bone fill (RBF) compared with FGF-2, no significant difference in clinical attachment level gain was observed. It was thus not possible to draw definite conclusions as to the true benefit of the combination therapy. Thus, further basic research addressing its effects was deemed necessary.

This study investigated the effects of the combined use of FGF-2 with DBBM on periodontal healing using in vivo and in vitro approaches.

## 2. Materials and Methods

### 2.1. Animals

Male Wistar rats (in vivo: 250–300 g, 10 weeks old; in vitro: 80–90 g, 4 weeks old) were purchased from Sankyo Labo Service (Tokyo, Japan). Animals were housed in individual cages maintained under standard laboratory conditions and fed a normal rat chow with water ad libitum. In vivo experimental protocol conformed to the Treatment of Experimental Animals at Tokyo Dental College (approval number 202203). This report adheres to the ARRIVE guidelines (https://www.nc3rs.org.uk/arrive-guidelines, accessed on 6 May 2021).

### 2.2. Biomolecules and Materials

Recombinant FGF-2 was provided by Kaken Pharmaceutical Co., Ltd. (Tokyo, Japan; Lot No. 90 HCC). DBBM with a particle size of 0.25–1.0 mm (Geistlich Bio-Oss^®^, Geistlich Pharma AG, Wolhusen, Switzerland) was used. For in vivo experiments, FGF-2 was diluted with distilled water and then mixed with 3% hydroxypropyl cellulose (HPC) (FUJIFILM Wako Pure Chemical, Osaka, Japan) to produce a 0.3% FGF-2 solution. This concentration was based on that used in clinical treatment. In order to adjust particle size for filling the surgical defect in vivo, DBBM was loosely pulverized with a sterile mortar and pestle as described previously [22]. For the FGF-2 + DBBM group, DBBM was pre-mixed with FGF-2 and used after 10 min.

### 2.3. In Vivo Experimental Design and Surgical Procedures

Animals were assigned to four subgroups: (1) Unfilled (*n* = 10), (2) FGF-2 (*n* = 10), (3) DBBM (*n* = 10), and (4) FGF-2 + DBBM (*n* = 10) (Appendix A). They were anesthetized with intraperitoneal injection of the mixture of medetomidine (0.15 mg/kg), midazolam (2 mg/kg), and butorphanol (2.5 mg/kg), and local infiltration anesthesia (2% xylocaine, 1:80,000 adrenaline) was given. After crestal incision and full-thickness flap elevation, standardized periodontal defects (2.0 × 2.0 × 1.7 mm, width × length × depth) were created mesially of the maxillary first molars (M1) [23], using a surgical template (Appendix A) [24]. Surgery was performed using a magnification loupe (×4.3 EyeMag Pro; Carl Zeiss, Oberkochen, Germany) with a light source. M1 root was carefully denuded of its PDL, cementum, and superficial dentin. The defects were rinsed with saline and dried with sterile gauze and were given 30 µL of 0.3% FGF-2 mixed with HPC, 2 mg of DBBM, FGF-2 + DBBM, or left unfilled. The flaps were closed using resorbable sutures (Vicryl^®^ 6–0; Johnson & Johnson, Tokyo, Japan) (Appendix A). Acetaminophen was used for pain control.

### 2.4. Microcomputed Tomography Analysis

At 2 or 4 weeks postoperatively, cardiovascular perfusion was performed with 4% paraformaldehyde after the animals were anesthetized. Maxillae were retrieved, and the healing in the surgical defect was examined using a microcomputed tomography (micro-CT) system (R-mCT; Rigaku, Tokyo, Japan). The exposure conditions were 90 kV and 150 μA. The magnification was set at ×10, and a slice width was 16 µm. The information from all the slices was saved at 484 × 481 pixels. The region of interest (ROI) was defined by the following criteria: (1) longitudinally, from the level of the alveolar bone crest at the mesial root of M1 to a depth of 1.7 mm and (2) horizontally, the box-shaped defect that includes mesial aspect of the mesial root of M1 (distal limit) and a mesial extension of 2.0 mm (mesial limit) with the buccal and palatal aspects of the bone defect. The ROI was placed where the original defect was located, as the margins were visually recognizable. Image data obtained by micro-CT were analyzed by 3-D structural analysis software (TRI/3D-BON; Ratoc System Engineering, Tokyo, Japan).

For the assessment of bone mineral density (BMD), a calibration curve was generated according to the hydroxyapatite contents using an imaging phantom (Kyoto Kagaku, Kyoto, Japan) [25]. Within the ROI, newly formed bone was defined to have a BMD of 400–1000 mg/cm^3^, and the DBBM particles and existing bone were to have a BMD above 1000 mg/cm^3^, referencing the method described by Zhang, Ding, and Kasugai [26]. The region of newly formed bone was calculated by subtracting the defined region of DBBM and existing bone from the total radiopaque region of BMD above 400 mg/cm^3^.

Bone volume fraction (bone volume/total volume: BV/TV) within the ROI were then analyzed (excluding DBBM particles). In case of combination therapy using bone substitutes, the evaluation of new bone formation only may underestimate clinical success; the ratio of total radiopaque volume (newly formed bone + residual DBBM particles) to total volume (RV/TV) within the ROI was also evaluated [27].

### 2.5. Histological Analysis

The maxillae were divided into two parts along the palatal median line. The samples were demineralized in 10% ethylenediaminetetraacetic acid disodium salt (EDTA-2Na, pH 7.0) (Muto Pure Chemicals, Tokyo, Japan) at 4 °C for 3 weeks after fixation in buffered 4% paraformaldehyde for 24 h, and then embedded in paraffin. Mesio-distal sections (thickness 5 µm) were cut with a microtome (Hyrax M25; Carl Zeiss). From each specimen, five to ten sections representing the central portion of the root in the defect were stained with hematoxylin–eosin or Azan–Mallory staining for histological and histomorphometric analyses [28].

### 2.6. Histomorphometric Analysis

Histomorphometric analysis of epithelial down-growth was performed using a light microscope (UPM Axiophot 2; Carl Zeiss Japan, Tokyo, Japan) and software (Axio Vision 4.7; Carl Zeiss Japan) by the method of Bizenjima et al. [29].

On the root-planed surface of M1, the angulation of PDL-like fibers was analyzed by Image J software (https://imagej.nih.gov/ij/, accessed on 6 May 2021) according to the method of Park et al. [30].

### 2.7. Immunohistochemistry

The sections were incubated in 3% hydrogen peroxide with methanol for 30 min at room temperature after deparaffinization. The sections were heated in a microwave for 3 min for antigen retrieval. After washing with phosphate-buffered saline (PBS, pH 7.4), the sections were incubated with 3% bovine serum albumin for 30 min to block nonspecific binding.

For the analysis of proliferating cell nuclear antigen (PCNA), the sections were incubated for 12 h at 4 °C with mouse anti-PCNA monoclonal primary antibody (diluted 1:100) (PC-10; Agilent Technologies, Santa Clara, CA, USA). Then sections were incubated with horseradish peroxidase (HRP)-conjugated secondary antibody (Histofine^®^ Simple Stain Max PO (M); Nichirei Biosciences, Tokyo, Japan) for 60 min at room temperature. They were rinsed with PBS, incubated with diaminobenzidine (Histofine^®^ Simple Stain DAB; Nichirei Biosciences, Tokyo, Japan), and counterstained with hematoxylin.

Detections of vascular endothelial growth factor (VEGF) and osterix (Osx) were carried out using anti-VEGF monoclonal antibodies (diluted 1:50) (Abcam, Cambridge, UK) and anti-Osx polyclonal antibodies (diluted 1:500) (Abcam), respectively.

For the observation of PCNA-, VEGF-, and Osx-positive cells, magnification was set at ×200, and a field of connective tissue (350 μm × 470 μm area) was randomly sampled in each section and submitted to quantitative analysis using an image analyzer (Image-Pro Plus 6.2, Media Cybernetics, Rockville, MD, USA), essentially as described previously [31]. On each specimen, analysis sites were compartmentalized into three areas (Root side, Bone side, and Middle area) to count PCNA-, VEGF-, or Osx-positive cells [24,32].

The micro-CT, histomorphometric, and immunohistochemical data were evaluated by one examiner who was blinded to the experimental grouping and confirmed by a second independent examiner. These examiners were trained and calibrated prior to the experiment.

### 2.8. In Vitro Cell Culture

For evaluation of the effects of FGF-2 and/or DBBM on cell viability/proliferation and morphology, PDLCs were isolated from the upper incisors of 4-week-old, male Wistar rats as previously described [33]. MC3T3-E1 cells were purchased from RIKEN BioResource Center (Ibaraki, Japan) and used for the analysis of osteoblast differentiation and mineralization.

Cells were incubated in α-minimal essential medium (αMEM, Gibco, Invitrogen, Carlsbad, CA, USA), supplemented with heat-inactivated 10% fetal bovine serum and antimicrobials at 37 °C in 5% CO_2_ in air.

To minimize possible changes in medium composition due to ion leaching [34], DBBM was immersed in the medium 5 days before the start of the experiment. For the evaluation of FGF-2 treated DBBM, after aspirating the medium completely, the FGF-2 solution (5 µg/mL) was mixed with DBBM. Samples were maintained for 10 min at room temperature, and then the media containing cells were added.

### 2.9. Enzyme-Linked Immunosorbent Assay and Confocal Laser Scanning Microscopy

Release of FGF-2 from DBBM was measured by enzyme-linked immunosorbent assay (ELISA). A total of 100 µL of the FGF-2 solution was mixed with 100 mg of DBBM and maintained for 10 min at room temperature. The samples were washed in PBS and vortexed twice to remove non-absorbed FGF-2, according to the method described previously [35]. The samples were immersed in 500 µL of PBS in a 24-well plate. Supernatant was collected, replaced with fresh PBS at set timepoints, and stored at −80 °C. The amount of FGF-2 was measured using an FGF-2 ELISA kit (R&D System, Minneapolis, MN, USA) according to the manufacturer’s instructions. The assay sensitivity was less than 3 pg/mL. FGF-2 concentrations were calculated from the standard curve generated.

Assessment of adsorption of FGF-2 onto DBBM surfaces was performed by confocal laser scanning microscopy (CLSM) in reference to the method described by Huh et al. [36]. At predetermined timepoints, the samples were fixed with 4% paraformaldehyde for 20 min and incubated with 10% goat serum (FUJIFILM Wako Pure Chemical Corporation, Osaka, Japan) for 60 min at room temperature, and then an Alexa Fluor 488-conjugated anti FGF-2 antibody (1:100, sc-365106, Santa Cruz Biotechnology, CA, USA) was applied to the samples. The specimens were imaged with CLSM (LSM880, Carl Zeiss). The magnification was set at ×100, and a series of Z-stack images were scanned in 2 μm increments using excitation wavelengths of 488 nm. The maximum projection of each stack was obtained using ZEN 2 black software (Carl Zeiss).

### 2.10. Cell Viability/Proliferation Assay

In order to assess the effect of adding FGF-2 to DBBM on cell viability/proliferation, PDLCs were plated at a density of 1 × 10^4^ cells/well in 24-well plates containing culture media with 100 mg of DBBM mixed with/without 100 µL of FGF-2. At set timepoints, WST-1 assay (Takara Bio, Otsu, Japan) was performed according to the manufacturer’s protocol.

### 2.11. Assessment of Cell Morphology and Spreading

Cell morphology and spreading were assessed by scanning electron microscopy (SEM) and CLSM. For SEM, PDLCs were seeded on 100 mg of DBBM mixed with/without 100 µL of FGF-2 at a density of 2 × 10^5^ cells/well in a 24-well culture plate. For SEM analysis, after 24 h of seeding, samples were rinsed twice with PBS and fixed with 2.5% glutaraldehyde in 0.1 M sodium cacodylate buffer (pH 7.4) and dried to the critical point of t-butyl alcohol, and then coated with Au-Pd using an SC500A sputter coater (Bio-Rad, Hercules, CA, USA). Cell morphology was observed with SEM (SU6600; Hitachi High-Tech Corporation, Tokyo, Japan).

For CLSM, after 24 h, the samples were fixed as described in the adsorption assay. After blocking and permeabilizing with 0.1% Triton X-100 for 5 min at room temperature, the actin cytoskeleton was stained using Alexa Fluor-488 Phalloidin (diluted 1:100) (Life Technologies, Carlsbad, CA, USA) and cell nuclei were counterstained with 4′,6-diamidino-2-phenylindole (DAPI) (diluted 1:1000) (Thermo Scientific, Pittsburgh, PA, USA). The cells were examined under a CLSM (Carl Zeiss). The specimens were imaged with CLSM as described in the adsorption assay, using the excitation wavelengths of 405 and 488 nm.

### 2.12. Alkaline Phosphatase (ALP) and Alizarin Red Staining

ALP staining was performed using Leukocyte ALP kit (Sigma-Aldrich, MO, USA). MC3T3-E1 cells were plated at a density of 5 × 10^4^ cells/well in 24-well plates with 8 mg of DBBM with/without 8 µL of FGF-2. The samples were rinsed twice with PBS and fixed with 4% paraformaldehyde for 20 min after 7 days. An alkaline dye solution was prepared by mixing sodium nitrite solution, fast red violet alkaline solution, deionized water, and naphthol AS-BI alkaline solution. The solution was added to the samples and incubated at room temperature for 15 min. After incubation, the samples were rinsed in deionized water.

The extracellular matrix mineralization was evaluated by alizarin red staining (Alizarin Red S; Sigma-Aldrich) as described previously [37]. After 14 days of seeding MC3T3-E1 cells, the samples were rinsed twice with PBS and fixed with 96% ethanol for 15 min and stained with 0.2% alizarin red solution in water (pH 6.4) at room temperature for 1 h.

For ALP and alizarin red staining, samples were also incubated without cells and served as “no cell” controls. Images were captured on a microscope camera (Zeiss Stemi 508 with AXIOCAM ERc 5s camera; Carl Zeiss) and imported onto Image J. Color thresholding was used to determine the percent stained values for each field view, according to the method by Fujioka-Kobayashi et al. [37].

### 2.13. Statistical Analysis

The sample size was determined by a power analysis based on a 90% power with a 0.05 two-sided significance level, given a 10.5% difference in bone volume between groups (as measured by micro-CT) and a standard deviation of 7% [29]. According to the determination, a sample size of 10 (defect sites) in each group was needed. Considering the potential loss of animals or samples during experiments, we included 12 rats per group.

Inter-group comparisons for the micro-CT and histomorphometric analyses were made by analysis of variance (ANOVA) with Tukey post hoc test. Inter- and intra-group comparisons for immunohistochemical analyses were performed by Kruskal–Wallis test with Dunn’s post hoc test. Inter-group comparisons for the WST-1 assay, ALP, and alizarin red staining data were performed by Mann–Whitney U test, and intra-group comparisons for the WST-1 assay were made by Friedman test with post hoc test. A software package (Prism ver 7.05; GraphPad Software, San Diego, CA, USA) was used. A *p*-value less than 0.05 was considered statistically significant.

## 3. Results

### 3.1. Micro-CT Analysis

Six rats died during surgery; all other animals had an uneventful recovery. Postoperative healing progressed normally. Micro-CT images at 2 weeks showed limited new bone formation in the Unfilled group (Figure 1a). Obvious bone formation was observed near the root in the FGF-2 group. In the DBBM and FGF-2 + DBBM groups, DBBM particles and new bone structure were observed adjacent to the root.

BV/TV values in the FGF-2 and FGF-2 + DBBM groups were significantly greater compared with the Unfilled or DBBM groups (Figure 1b).

At 4 weeks, new bone was still limited in the Unfilled group, but it appeared to be enhanced in other groups (Figure 1d). FGF-2, DBBM, and FGF-2 + DBBM groups showed significantly greater BV/TV values compared with the Unfilled group (Figure 1e). Significantly greater values were observed in the FGF-2 and FGF-2 + DBBM groups compared with the DBBM group. There was no significant difference in BV/TVs between the FGF-2 and FGF-2 + DBBM groups at either time point.

At 2 and 4 weeks, RV/TV values in the DBBM and FGF-2 + DBBM groups were significantly greater than those in the Unfilled or FGF-2 groups (Figure 1c,f).

### 3.2. Histological Analysis

At 2 weeks, new bone was observed around the Root side in all groups (Figure 2a–d). The previous defect was filled with connective tissue in all groups. The DBBM particles were observed in the DBBM and FGF-2 + DBBM groups.

At 4 weeks, in the FGF-2 (Figure 2f), DBBM (Figure 2g), and FGF-2 + DBBM (Figure 2h) groups, bone formation has progressed along the root. In the Unfilled group, it was still limited (Figure 2e). The extent of bone formation in the FGF-2 and FGF-2 + DBBM groups appeared to be greater than that in the DBBM group.

Figure 2i,j shows the enlarged view of framed area in the corresponding group at 4 weeks. In the DBBM group (Figure 2i), graft particles were encapsulated by fibrous connective tissue with mild inflammatory infiltrates, whereas in the FGF-2 + DBBM group (Figure 2j), particles located near the Root side of the defect were incorporated in new bone.

There were no signs of ankylosis in any group during the experiment.

### 3.3. Histomorphometric Analysis

Epithelial down-growth in the FGF-2 and FGF-2 + DBBM groups appeared to be less than that in the Unfilled and DBBM groups at 2 and 4 weeks (Figure 3a–h). The results of the quantitative analysis supported such observations (Figure 3i,j).

Photomicrographs of samples stained by Azan–Mallory staining are shown in Appendix A. At 2 weeks, directions of PDL-like collagen bundles were observed generally along root surfaces in all groups (Appendix A). At 4 weeks, in the FGF-2 and FGF-2 + DBBM groups, the fiber bundles ran obliquely to the root surface (Appendix A). Newly formed cementum-like structure was observed in the FGF-2 and FGF-2 + DBBM groups.

In quantitative analysis of 2-week samples, there was no significant difference in values for fiber angulation between groups (Appendix A). At 4 weeks, the values in the FGF-2 and FGF-2 + DBBM groups were significantly greater compared with the Unfilled group (Appendix A).

### 3.4. Immunohistochemical Analyses

At 2 weeks, in the Root side (Appendix A) and Bone side (Appendix A), the prevalence of PCNA-positive cells in the FGF-2 and FGF-2 + DBBM groups appeared to be higher than that in the Unfilled group. In the Middle area (Appendix A), many PCNA-positive cells were observed around DBBM particles in the DBBM and FGF-2 + DBBM groups. The results from quantitative analysis were consistent with these observations (Table 1). At 4 weeks, no significant difference was observed among the groups in all areas (Appendix A, Table 1).

VEGF-positive cells were often observed near blood vessels in connective tissue and around newly formed bone (Appendix A). At 2 weeks, in all areas, VEGF-positive cells appeared to be increased more in the FGF-2, DBBM, and FGF-2 + DBBM groups than in the Unfilled group (Appendix A). In quantitative analysis, the proportions of VEGF-positive cells in the FGF-2 and FGF-2 + DBBM groups were significantly greater than in the Unfilled group (Table 1). In the Middle area (Appendix A), the proportions of VEGF-positive cells in the DBBM group were significantly greater than that in the Unfilled group (*p* < 0.05). At 4 weeks, no significant difference was observed among the groups in all areas (Appendix A, Table 1).

Osx-positive cells were often observed around newly formed bone and DBBM particles (Figure 4a–f). At 2 weeks, in the FGF-2 and FGF-2 + DBBM groups, Osx-positive cells appeared to be increased in both the Root side (Figure 4a) and Bone side (Figure 4b). In the Middle area (Figure 4c), almost no Osx-positive cells were observed in the Unfilled and FGF-2 groups. The results from quantitative analysis supported these observations (Table 1). At 2 and 4 weeks, in the Middle area (Figure 4c,f), the proportions of Osx-positive cells in the DBBM and FGF-2 + DBBM groups were significantly greater compared with the Unfilled and FGF-2 group (Table 1).

### 3.5. Release Kinetics of FGF-2

When the FGF-2 release kinetics from the FGF-2-treated DBBM was assessed by ELISA, a rapid release, approximately 67% of the total amount, was confirmed by 12 h (Figure 5a). After that, slower FGF-2 release was noted over 120 h.

CLSM images demonstrated the adsorption of FGF-2 to the DBBM surface up to 120 h (Figure 5b–e).

### 3.6. Cell Viability/Proliferation

PDLCs on the FGF-2-treated DBBM showed significantly higher viability/proliferation than on non-treated DBBM (Appendix A). In intra-group comparisons, PDLCs on the FGF-2-treated DBBM demonstrated a significant increase in viability/proliferation at 72 and 120 h.

### 3.7. Cell Morphology and Behavior

SEM images showed that PDLCs attached to the DBBM and FGF-2-treated DBBM (Figure 6a–d). Compared with non-treated DBBM (Figure 6a,c), more cells appeared to be attached to the surface of the FGF-2-treated DBBM (Figure 6b,d). Lamellipodia-like cell protrusions were more frequently observed on the FGF-2-treated DBBM (Figure 6b,d).

Similarly, CLSM images showed a greater number of cells adhered to and spread on the FGF-2-treated DBBM (Figure 6f) compared with DBBM (Figure 6e).

### 3.8. ALP Activity and Alizarin Red Staining

Compared with cells on non-treated DBBM (Figure 7a), ALP activity of MC3T3-E1 cells on the FGF-2-treated DBBM (Figure 7b) appeared to increased. The quantification results (Figure 7c) supported such observation.

Alizarin red staining images of DBBM (Figure 7d) showed dark coloration similar to those of “no cell” control (not shown). FGF-2-treated DBBM demonstrated significantly greater alizarin red staining (Figure 7e). Quantification data (Figure 7f) showed that staining level was significantly greater on the FGF-2-treated DBBM compared with DBBM.

## 4. Discussion

This study investigated the effects and mechanisms of healing following the use of FGF-2 and DBBM. Our micro-CT and histological data indicate that the combination therapy using FGF-2 and DBBM was similarly effective as FGF-2 alone, in the healing of experimental periodontal defects. The immunohistochemical and in vitro data suggest that adding FGF-2 to DBBM may enhance healing via promotion of cell proliferation, angiogenesis, and osteogenic differentiation.

In micro-CT analysis, BV/TV values in the FGF-2 and FGF-2 + DBBM groups were significantly greater than in the Unfilled and DBBM groups. The results were consistent with the histological observations. In a previous study, when DBBM was used to treat rabbit calvarial defects, graft particles located near the defect margin were integrated into newly formed mineralized bone, but those in the central area were surrounded by fibrous connective tissue at 2 weeks [38]. By 4 weeks, formation of bony islands was confirmed. These data support the osteoconductive property of DBBM. In the FGF-2 + DBBM group of the current study, particles located near the Root side were often incorporated in newly formed bone at 4 weeks. These collectively suggest that adding FGF-2 to DBBM promotes bone healing. It is of interest to note that, in comparison between the FGF-2 + DBBM and FGF-2 alone groups, no significant difference was found in the micro-CT and histological results. This finding was in line with our clinical observations [12,21].

As for combination therapy using bone substitutes, micro-CT analysis of bone formation could underestimate its effect, because the presence of the residual particles can limit the volume data of new bone within the ROI. In clinical studies, RBF (i.e., new bone and the remnants of bone substitutes) is often used as a measure for clinical outcome [12]. Therefore, we also evaluated the total radiopaque volume (new bone + DBBM) per total volume (RV/TV) [27]. RV/TVs were significantly greater in the DBBM and FGF-2 + DBBM groups than in the Unfilled and FGF-2 groups. In histological analysis, bone formation in the DBBM and FGF-2 + DBBM groups at 4 weeks appeared to be greater compared with that at 2 weeks. In pre-clinical studies, application of DBBM to calvarial bone defects enhanced bone formation [39,40]. These results suggest that the increase in bone levels in the DBBM and FGF-2 + DBBM groups was due not only to the presence of DBBM but also to novel bone formation.

At 2 weeks, PCNA-, VEGF-, and Osx-positive cells were prominently observed in the Root side in the FGF-2 and FGF-2 + DBBM groups. VEGF is a factor that induces the proliferation of endothelial cells and stimulates angiogenesis [41]. In a periodontal defect model of dog, FGF-2 has been shown to promote cell proliferation, angiogenesis, and the expression of osteoblast differentiation markers at 1 week postoperatively [8]. In the present study, most of the proliferated cells in the Root side were considered to be derived from PDL. Moreover, epithelial down-growth was limited in the FGF-2 and FGF-2 + DBBM groups. In vitro, FGF-2-induced proliferation of gingival epithelial cells was inhibited in the presence of serum [42]. In the FGF-2 and FGF-2 + DBBM groups, the formation of a thin layer of cementum-like structure was frequently observed, and the PDL-like collagen bundles were oriented oblique to the root surface. We speculate that FGF-2 enhanced the formation of new connective tissue attachment via promoting the PDLCs in the Root side in the FGF-2 and FGF-2 + DBBM groups.

Interestingly, in the Middle area, the DBBM and FGF-2 + DBBM groups showed significantly more Osx-positive cells, especially around DBBM particles, than the Unfilled and FGF-2 groups. Osx is one of the important transcription factors in osteoblast differentiation [43]. With reference to our previous study using the same defect model [29], it is considered that the healing process in the Unfilled and FGF-2 groups is initiated with the proliferation and differentiation of mesenchymal stem cells (MSCs) near the root and existing bone, and later spreads to the Middle area. In the current study, some DBBM particles were surrounded by connective tissue and fibroblast-like cells in both the DBBM and FGF-2 + DBBM groups. These cells contain MSCs that could differentiate into osteoblasts and cementoblasts [8,44]. In the previous studies, DBBM induced a high vascularization in mouse connective tissue at 10 days after implantation [45], and the gene expression of Osx in mouse periodontal defects at 3 months [46]. In the current study, in all areas, the proportions of PCNA-, VEGF-, and Osx-positive cells in the FGF-2 + DBBM group were significantly greater than that in the Unfilled group at 2 and 4 weeks. It is considered that FGF-2 facilitated cell proliferation in the Root side and Bone side, and around DBBM particles in the Middle area, which consequently increased angiogenesis and osteoblast differentiation.

In vitro, PDLCs on the FGF-2-treated DBBM showed significantly higher viability/proliferation than non-treated DBBM. At 24 h, FGF-2-treated DBBM displayed greater numbers of PDLCs than DBBM. Cells on the FGF-2-treated DBBM frequently showed lamellipodia-like protrusions. In vitro studies reported that FGF-2 promoted cell proliferation via increasing the expression of CD44, a cell surface molecule of PDLCs involved in adherence and proliferation [4,47]. Generally, elongated pseudopodia of cells indicate good adhesion to the substrate [48]. These findings indicate that FGF-2 promotes early cell adhesion and spreading on DBBM, which contributes to the enhanced cell proliferation.

FGF-2 promoted ALP activity and alizarin red staining of MC3T3-E1 cells, when cultured with DBBM. A previous study reported that the addition of FGF-2 to Si-doped hydroxyapatite enhanced gene expression of osteoblast differentiation markers and matrix calcification in MC3T3-E1cells [49]. The results indicated that FGF-2 induced osteoblast differentiation and matrix calcification of cells on DBBM.

FGF-2 has a short half-life in vivo [50]. In a mouse fracture model, only 9% of the injected FGF-2 remained at the site at 3 days after injection, although endogenous FGF-2 was detected over 3 weeks [51]. In the treatment of mouse calvarial defect with a barrier membrane loaded with FGF-2, the peak release of FGF-2 during the first 72 h was thought to be responsible for granulation tissue formation and angiogenesis [52]. In the current study, FGF-2 release from and its adsorption to DBBM were observed over 120 h. Because DBBM is a deproteinized matrix, it easily adsorbs new proteins [53]. The adsorption of proteins to biomaterials and sustained release are the determining factors influencing downstream cellular behaviors [54,55]. The enhanced cell proliferation and osteoblast differentiation observed may be ascribed to the sustained release of FGF-2 from DBBM.

As already mentioned, the micro-CT analysis of bone formation showed no significant difference between the FGF-2 and FGF-2 + DBBM groups. Combined use of signaling molecules and bone substitutes is clinically recommended for the treatment of deep non-contained intrabony defects [56,57]. In our recent clinical study [21], the treatment of intrabony defects with FGF-2 + DBBM yielded significantly greater RBF when compared with FGF-2 alone at 2 years postoperatively, in cases of 1–2-wall defects. This presented the possibility that such combination therapy may be clinically beneficial in certain bone defect configurations. Our immunohistochemical (Osx) and in vitro data support the implication. Although the standardized defect in this study was relatively large, it is necessary to compare the performance of FGF-2 therapy with and without DBBM, using a model with more challenging defect configuration.

There are some relevant limitations to this study. Because of the small animal size and the type of defect model used, it is difficult to assess the extent of periodontal regeneration (e.g., cementum formation) in detail. In this study, we used a xenogeneic bone substitute, which is widely used in clinical treatment. However, other biomaterials, including synthetic ones, need to be tested for their potential as a scaffold or carrier material for FGF-2.

## 5. Conclusions

Within the limitations of this study, the use of FGF-2 with DBBM was similarly effective as FGF-2 alone in the healing of experimental periodontal defects. It is suggested that the combined use of FGF-2 and DBBM stimulates cell proliferation, angiogenesis, and osteoblast differentiation, which in turn could enhance healing in certain bone defect configurations.

## Figures and Tables

**Figure 1 biomolecules-11-00805-f001:**
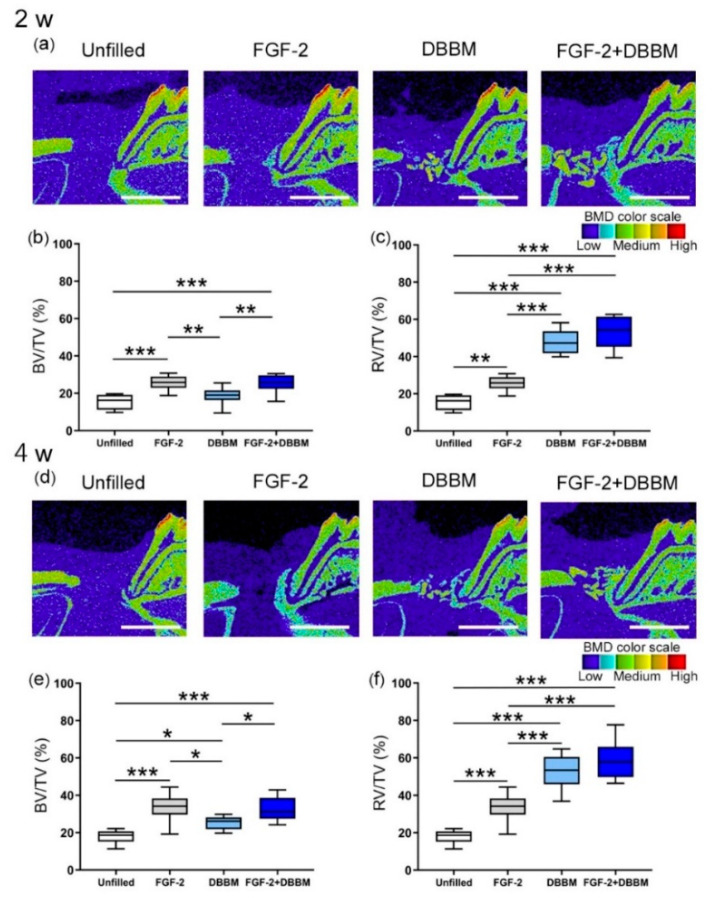
Two-dimensional, micro-CT images and quantitative analysis by 3-D structural analysis software (TRI/3D-BON). The color scale indicates: red and orange = high bone mineral density (BMD), yellow and green = medium BMD, and light blue and purple = low BMD. (**a**,**d**) Sagittal images from micro-CT at 2 and 4 weeks (bar = 2000 µm). Enhanced new bone formation can be observed in the FGF-2 and FGF-2 + DBBM groups. (**b**,**c**,**e**,**f**) Quantitative analysis of micro-CT images by a 3-D structural analysis software (TRI/3D-BON). Box in white, Unfilled group; box in gray, FGF-2 group; box in light blue, DBBM group; box in blue, FGF-2 + DBBM group. Bone volume (BV)/total volume (TV) within the ROI (**b**,**e**), radiopaque volume of newly formed bone and residual DBBM particles (RV)/total volume (TV) within the ROI (**c**,**f**) were compared between groups. Data shown as box-and-whiskers plot with minimum, maximum, median, and 25th and 75th percentiles (*n* = 10). * *p* < 0.05, ** *p* < 0.01, *** *p* < 0.001 by ANOVA with Tukey post-test.

**Figure 2 biomolecules-11-00805-f002:**
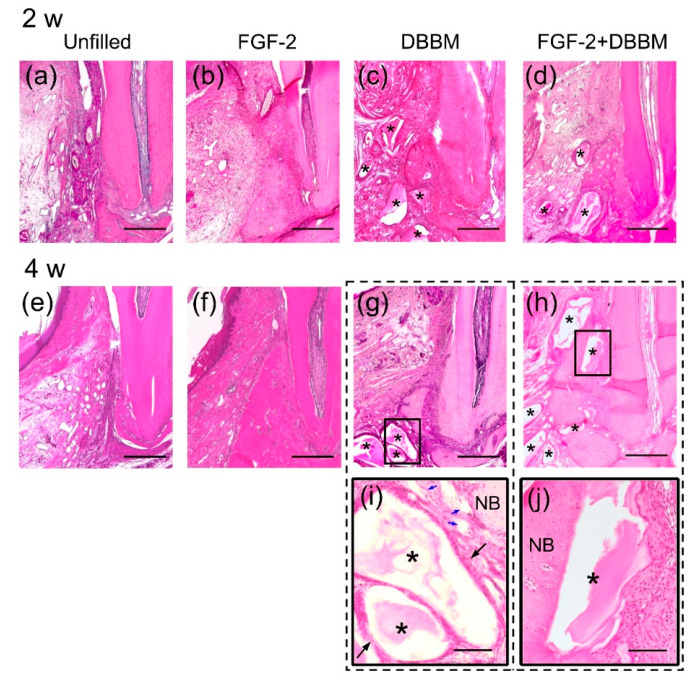
Histopathological overview (H&E staining). (**a**–**d**) Representative photomicrograph of each group at 2 weeks postoperatively (original magnification ×25, bar = 500 µm). Asterisk indicates DBBM particle. (**a**) New bone formation is limited in the Unfilled group. (**b**–**d**) In the FGF-2, DBBM, and FGF-2 + DBBM groups, newly formed bone can be observed from the Root side of the intrabony defect. (**e**–**h**) Representative photomicrograph at 4 weeks postoperatively (original magnification ×25, bar = 500 µm). (**e**) New bone formation is minimal in the Unfilled group. (**f**–**h**) The extent of newly formed bone in the FGF-2 and FGF-2 + DBBM groups appear to be greater than the DBBM and the Unfilled groups. (**i**,**j**) Higher magnification of framed area of the corresponding group at 4 weeks postoperatively (original magnification ×100; bar = 200 µm). In the DBBM group (**i**), DBBM particles are encapsulated by fibrous connective tissue (indicated by black arrows). The blood vessels in connective tissues around the DBBM particles are observed (indicated by blue arrows). In the FGF-2 + DBBM group (**j**), DBBM particles near the Root side of the previous defect are incorporated in the newly formed bone (NB).

**Figure 3 biomolecules-11-00805-f003:**
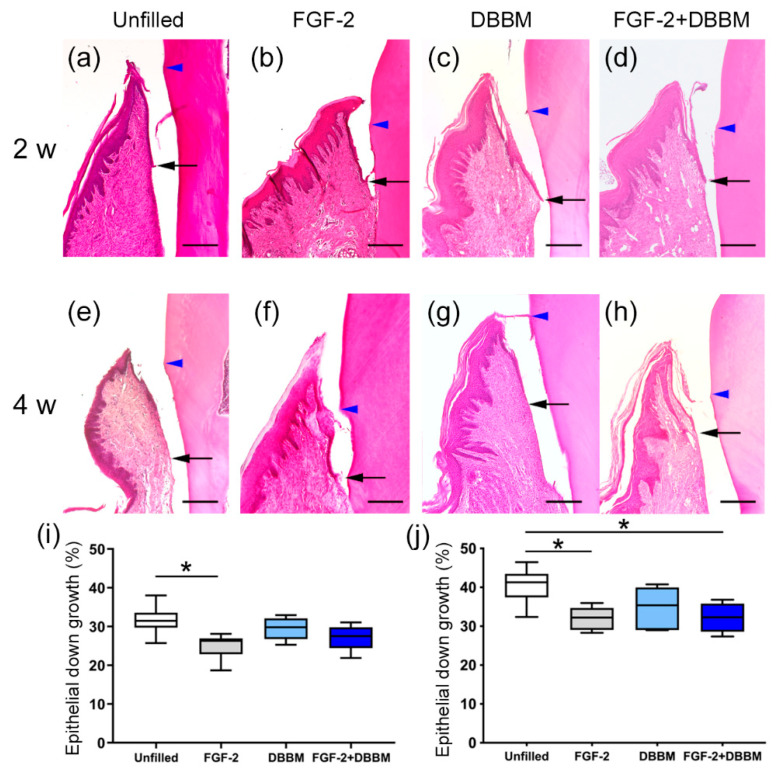
Histomorphometric assessment of epithelial down-growth. (**a**–**h**) The apical extent of epithelial down-growth is indicated by black arrow. Blue arrowhead indicates CEJ. (H&E staining) (**a**–**h**) At 2 and 4 weeks postoperatively, epithelial down-growth along the root surface in the FGF-2 and FGF-2 + DBBM groups appears to be less compared with the Unfilled and DBBM groups. (**e**–**h**) At 4 weeks postoperatively, a general trend for progression of the down-growth can be observed (original magnification ×50; bar = 200 μm). (**i**,**j**) Quantification of epithelial down-growth (%) at 2 and 4 weeks, calculated by: (the length between the most coronal and most apical extents of the junctional epithelium)/(the distance between CEJ and the defect base). Box in white, Unfilled group; box in gray, FGF-2 group; box in light blue, DBBM group; box in blue, FGF-2 + DBBM group. Data shown as box-and-whiskers plot with minimum, maximum, median, and 25th and 75th percentiles (*n* = 6). * *p* < 0.05 by ANOVA with Tukey post-test.

**Figure 4 biomolecules-11-00805-f004:**
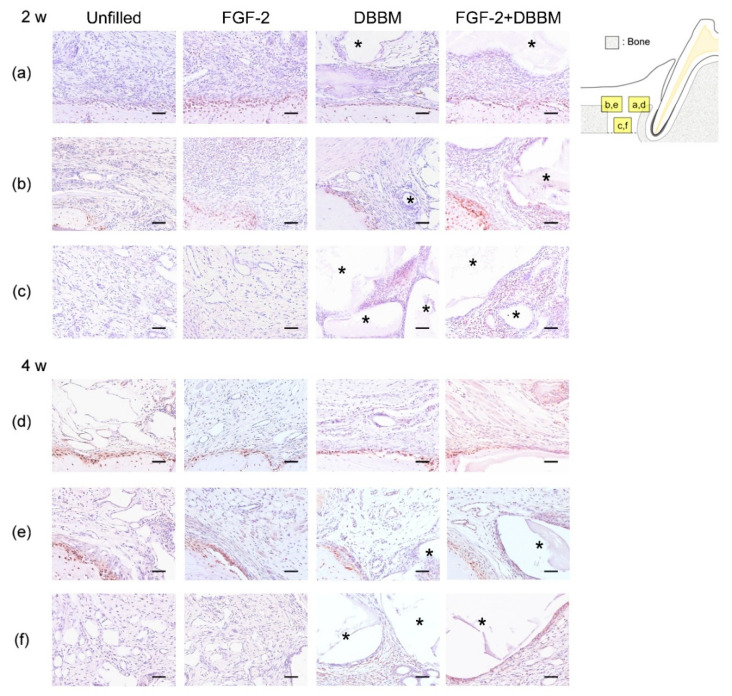
Representative photomicrographs of immunohistochemical staining for Osx. Prevalence of Osx-positive cells is assessed in the Root side (**a**,**d**), Bone side (**b**,**e**), and Middle area (**c**,**f**). A brown coloration indicates an Osx-positive reaction. In the Middle area at 2 and 4 weeks (**c**,**f**), Osx-positive cells are often observed around DBBM particles in the DBBM and FGF-2 + DBBM groups, whereas almost no Osx-positive cells are observed in the Unfilled and FGF-2 groups. (Osx and counterstaining with Mayer’s hematoxylin stain; original magnification ×200; bar = 50 µm; asterisk indicates DBBM particles).

**Figure 5 biomolecules-11-00805-f005:**
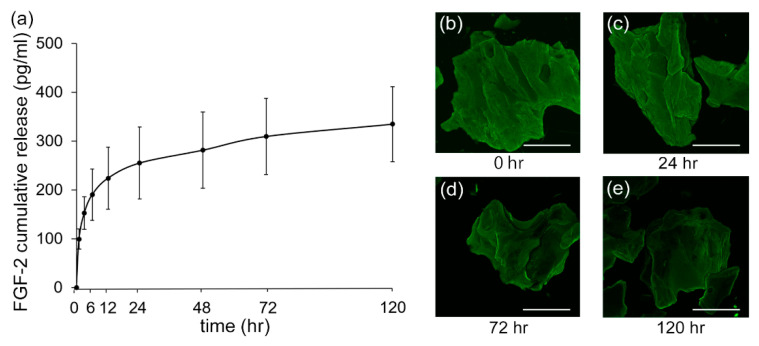
Kinetics of FGF-2 cumulative release from DBBM and adsorption of FGF-2 to DBBM. (**a**) FGF-2 release evaluated by ELISA. FGF-2 was released from DBBM treated with FGF-2 over 120 h. CLSM images of adsorption of FGF-2 to DBBM surfaces for (**b**) 0 h, (**c**) 24 h, (**d**) 72 h, and (**e**) 120 h. FGF-2 on DBBM is indicated by a green color (original magnification ×100; bar = 500 μm).

**Figure 6 biomolecules-11-00805-f006:**
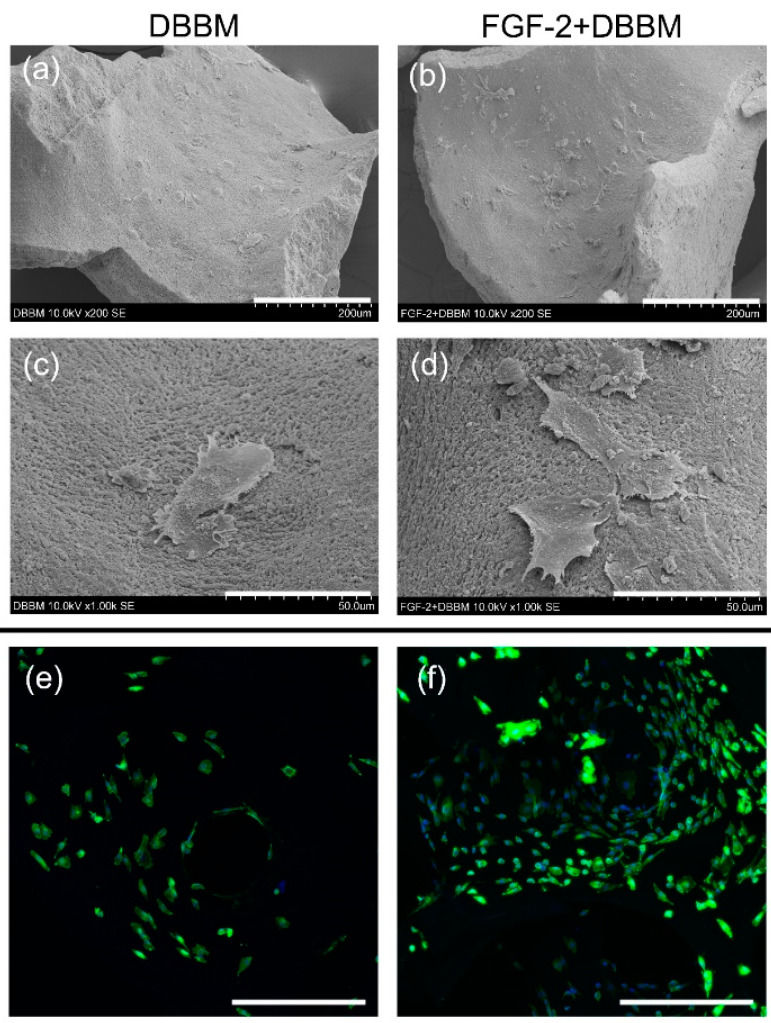
SEM and CLSM images of PDLCs on the DBBM with/without FGF-2. SEM images show that PDLCs attached to the DBBM (**a**,**c**) and DBBM treated with FGF-2 (**b**,**d**) at 24 h. (**a**,**b**) A greater number of cells appear to be attached to the surface of FGF-2-treated DBBM compared with DBBM. (**c**,**d**) Compared with cells on non-treated DBBM (**c**), cells with lamellipodia-like protrusions are more evident on the FGF-2-treated DBBM. ((**a**,**b**): original magnification ×200; bar = 200 μm, (**c**,**d**): original magnification ×1000; bar = 50 μm). (**e**,**f**) CLSM images show cells stained for actin (green) and the nucleus (blue) at 24 h. Compared with DBBM (**e**), a greater number of attached cells can be observed on the surface of FGF-2-treated DBBM (**f**). (original magnification ×100; bar = 500 μm).

**Figure 7 biomolecules-11-00805-f007:**
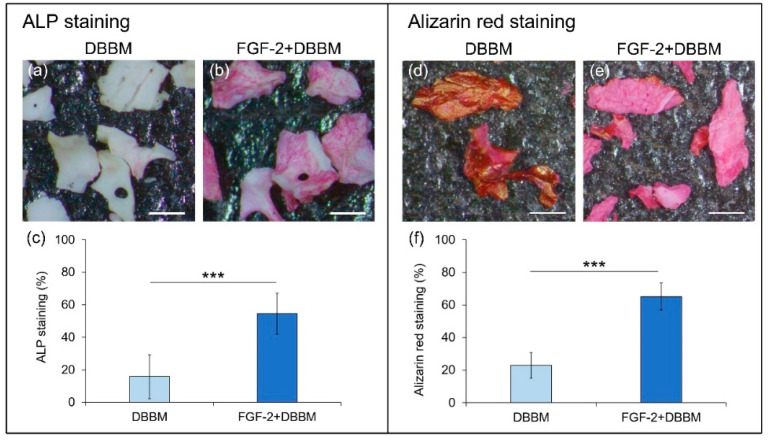
ALP staining and Alizarin red staining of MC3T3-E1 cells on DBBM with/without FGF-2. (**a**,**b**) Alkaline phosphatase staining on DBBM and FGF-2-treated DBBM at 7 days post-seeding (original magnification ×25; bar = 500 µm). Compared with DBBM (**a**), FGF-2-treated DBBM particles (**b**) show an increased level of ALP staining. (**c**) Quantification of ALP staining. (**d**,**e**) Visual representation of alizarin red stained particles on DBBM and FGF-2-treated DBBM at 14 days post-seeding (original magnification ×25; bar = 500 µm). Compared with DBBM (**d**), FGF-2-treated DBBM particles (**e**) show an increased alizarin red staining. (**f**) Quantification of alizarin red staining. Data shown as mean ± SD (*n* = 6) by Mann–Whitney U test, *** *p* < 0.001.

**Table 1 biomolecules-11-00805-t001:** Quantitative analysis of the PCNA-, VEGF-, and Osx-positive cells.

	Area/Group		Unfilled	FGF-2	DBBM	FGF-2 + DBBM
PCNA	2 w	Root side	20.5 ± 3.1 *	44.9 ± 9.7 ^a^	29.6 ± 6.6	46.1 ± 6.5 ^a^
		Bone side	18.6 ± 3.0	33.1 ± 5.8 ^a^	27.2 ± 5.0	34.5 ± 4.3 ^a^
		Middle area	15.7 ± 2.4	31.3 ± 7.0	37.3 ± 2.8 ^a,†^	39.3 ± 9.8 ^a^
	4 w	Root side	18.3 ± 7.0	16.2 ± 4.6	16.3 ± 4.7	17.6 ± 5.3
		Bone side	15.6 ± 5.2	14.1 ± 8.8	16.6 ± 4.6	13.6 ± 2.7
		Middle area	14.0 ± 3.1	10.1 ± 3.9	16.1 ± 4.1	16.9 ± 5.3
VEGF	2 w	Root side	7.7 ± 1.9	18.2 ± 3.7 ^a^	13.9 ± 4.6	19.0 ± 4.2 ^a^
		Bone side	6.7 ± 2.9	17.3 ± 3.0 ^a^	12.5 ± 5.0	18.0 ± 3.4 ^a^
		Middle area	7.1 ± 2.2	15.3 ± 3.6 ^a^	16.0 ± 4.0 ^a^	19.0 ± 6.1 ^a^
	4 w	Root side	8.8 ± 3.1	11.7 ± 5.3	9.9 ± 3.2	11.1 ± 3.5
		Bone side	8.3 ± 2.9	7.4 ± 3.3	9.5 ± 2.3	11.0 ± 3.6
		Middle area	9.2 ± 3.0	8.3 ± 4.6	12.3 ± 2.0	13.7 ± 3.4
Osx	2 w	Root side	8.4 ± 2.7 *	18.8 ± 5.4 ^a,^*	13.3 ± 3.6 *	19.6 ± 7.9 ^a,^*
		Bone side	4.8 ± 1.9	9.8 ± 2.7 ^a^	7.8 ± 1.4	9.4 ± 1.0 ^a^
		Middle area	0.2 ± 0.2	0.6 ± 0.8	7.1 ± 2.9 ^a,b^	8.0 ± 2.4 ^a,b^
	4 w	Root side	19.6 ± 5.9 *	14.6 ± 4.4 *	19.7 ± 4.6 *^,†^	17.2 ± 4.7 *
		Bone side	8.6 ± 5.2	8.8 ± 3.4	9.1 ± 4.1	9.4 ± 2.9
		Middle area	0.1 ± 0.2	0.6 ± 1.1	7.4 ± 4.2 ^a,b^	6.6 ± 2.3 ^a,b^

Data shown as mean ± SD (*n* = 6) of PCNA-, VEGF-, and Osx-positive cells/total cells (%) in the count area (Root side, Bone side, Middle area). ^a^, significantly different from the Unfilled group; ^b^, from the FGF-2 group. *, significantly different from the Middle area; †, from the Bone side; by Kruskal–Wallis test with Dunn’s post hoc test (*p* < 0.05–0.001).

## Data Availability

Supporting data for this study may be made available from the corresponding author upon reasonable request.

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
