# Peer review of "Healing of Experimental Periodontal Defects Following Treatment with Fibroblast Growth Factor-2 and Deproteinized Bovine Bone Mineral"

_biomolecules, 2021, doi:10.3390/biom11060805_

Round 1

Reviewer 1 Report

MANUSCRIPT: Healing of experimental periodontal defects following treatment with fibroblast growth factor-2 and deproteinized bovine bone mineral by Murakami and colleagues.

GENERAL COMMENTS: This pre-clinical study evaluated the effect of FGF-2 on periodontal defects as well as the osteogenic differentiation using cell cultures. FGF-2 was also associated with xenograft material to enhance the osteoconductive properties. Both in vivo and in vitro results showed that FGF-2 associated with xenograft material was similar with FGF-2 alone to treat periodontal defects. However, FGF-2 associated with xenograft material enhanced wound healing. The study was well designed and conducted depicting very interesting results. The manuscript deserves publication; however, some minor revision must be done before acceptance. Specific comments were made below.

SPECIFIC COMMENTS:

M&M

L73- The sample size was based on a tomographic variable as described in the L232. However, 6 animals were lost. Could the authors please clarify this issue? The sample size was enough??

L150: Did the examiners trained? Calibrated?

DISCUSSION

- The discussion should include the modulation of FGF-2 over the xenograft material. Did the authors speculate that DBBM is the only biomaterial available to carry the growth factor?

- The periodontal healing is based on periodontal ligament more than bone formation. Please, clarify.

- Please add the limitations of the study: sample size, biomaterial used, periodontal regeneration.

REFERENCES:  A total of 57 references; all pertinent.

Author Response

The study was well designed and conducted depicting very interesting results. The manuscript deserves publication; however, some minor revision must be done before acceptance. Specific comments were made below.

(Response)

We appreciate the positive comments. Please see our point-by-point responses to the review comments.

SPECIFIC COMMENTS:

M&M

L73- The sample size was based on a tomographic variable as described in the L232. However, 6 animals were lost. Could the authors please clarify this issue? The sample size was enough??

(Response)

The description of the sample size estimation was insufficient. We have considered potential dropouts. We added such information in the method section.

L150: Did the examiners trained? Calibrated?

(Response)

These examiners have been trained and calibrated. We added such information to the revised manuscript.

DISCUSSION

- The discussion should include the modulation of FGF-2 over the xenograft material. Did the authors speculate that DBBM is the only biomaterial available to carry the growth factor?

(Response)

There are other biomaterials available that may be used with FGF-2. As one example, we mentioned the use of FGF-2 with beta TCP in the Introduction section. Considering the reviewer’s comment, we described DBBM as a xenogeneic material in the Discussion section of the revised manuscript.

- The periodontal healing is based on periodontal ligament more than bone formation. Please, clarify.

(Response)

We completely agree with the reviewer’s view. We focused on the bone formation because it is one of the challenges of the currently available regenerative approaches. As for the healing of periodontal ligament, we provided Azan staining data in the supplemental file. We also decided to present the figure for epithelial downgrowth in the main text, not as a supplemental file, according to the advice of another reviewer.

  In light of the reviewer’s comment, we mentioned this as a study limitation.

- Please add the limitations of the study: sample size, biomaterial used, periodontal regeneration.

(Response)

We appreciate this important comment. We added our descriptions about the study limitations.

Reviewer 2 Report

This article describes the effects of FGF-2 and deproteinized bovine bone mineral on healing of experimental periodontal defects in vivo and in vitro. I think the results in this manuscript are of interest, particularly in terms of the data demonstrating the beneficial effects of the combination treatment of them in rats. The data are sound. However, there are few concerns that should be addressed.

1. The sentences in the lines 17 to 20 in the Abstract are necessary because these addressed experimental methods.

2. Two sentences in the lines 25 to 28 in the Abstract section and 485 to 488 in the Discussion section are not logical. In the front sentence, combination treatment of FGF-2 with DBBM was described to show a similar effect compared to FGF-2 single treatment in the healing of experimental periodontal defects. However, in the following sentence, authors emphasized the beneficial effects of FGF-2 on enhancing the healing. These two sentences should be revised.

3. As mentioned in the Introduction section, the authors’ previous randomized controlled trial study revealed that no significant differences were observed in clinical attachment level between co-treatment of DBBM and FGF-2 and DBBM alone (Reference 21). Authors should discuss the relation of that study to this study in the Discussion section. Furthermore, authors should discuss what is key findings in this study.

Author Response

I think the results in this manuscript are of interest, particularly in terms of the data demonstrating the beneficial effects of the combination treatment of them in rats. The data are sound. However, there are few concerns that should be addressed.

(Response)

Thank you for the positive comment. We tried to address the concerns to the best of our abilities.

  1. The sentences in the lines 17 to 20 in the Abstract are necessary because these addressed experimental methods.

(Response)

Did you mean “…are unnecessary?” We feel that such information is necessary because the author guide specifies that we include method information is the Abstract section.

  1. Two sentences in the lines 25 to 28 in the Abstract section and 485 to 488 in the Discussion section are not logical. In the front sentence, combination treatment of FGF-2 with DBBM was described to show a similar effect compared to FGF-2 single treatment in the healing of experimental periodontal defects. However, in the following sentence, authors emphasized the beneficial effects of FGF-2 on enhancing the healing. These two sentences should be revised.

(Response)

We thank the reviewer for this important comment. We completely agree with the reviewer. We decided to revise the conclusion.

  1. As mentioned in the Introduction section, the authors’ previous randomized controlled trial study revealed that no significant differences were observed in clinical attachment level between co-treatment of DBBM and FGF-2 and DBBM alone (Reference 21). Authors should discuss the relation of that study to this study in the Discussion section.

(Response)

We appreciate this important comment. Please understand that, in the previous clinical studies (Ref. 12, 21), we did not compare DBBM + FGF-2 with DBBM alone. DBBM alone group was not included in the clinical study. Rather, we compared DBBM + FGF-2 with FGF-2 alone. However, we agreed that discussion in relation to our clinical studies should be added. Considering the reviewer’s advice, we added such discussion.

Furthermore, authors should discuss what is key findings in this study.

(Response)

We added the summary of key findings in the beginning of the Discussion section.

Reviewer 3 Report

Some informations are lacking to the materials and methods section

  • justification of the concentration of FGF2 used
  • Histomorphometric analysis: how many sections have been assessed. How many evaluators?
  • Immunohistochemistry : representative images 3 areas of the should be added
  • What is the rationale for using an osteoblast cell line (MC3T3-E1) : have primary OB cultures been considered? Are the results similar?
  • what is the hypothesis to explain the fixation of FGF2 on DBBM?
  • ELISA : please add the sensitivity of the kit

In the results:

  • the supplementary data 2 are very important and should be included in the main text. How is the quality of the attachement (anchorage of fibers)? The use of polarized light should be considered (for example: Lallam-Laroye C et al, J Biomed Mater Res, 2006)
  • the quantification of the histological results have to be done (and not only illustrated using pictures). For Figure 2, staining for quantification of mineralized bone (Von Kossa?) and osteoid matrix will be very interesting.
  • in 3.7: what are the results on M3C3T1?
  • in 3.8: on what cells were these results obtained?  (results on both cell types are interesting)

Author Response

Some informations are lacking to the materials and methods section

justification of the concentration of FGF2 used

(Response)

The concentration of FGF-2 is based on the concentration used in clinical treatment. This information is added to the method section.

Histomorphometric analysis: how many sections have been assessed. How many evaluators?

(Response)

The information on the number of sections was added to the revised manuscript.

As for the evaluators, such information was already provided in the last sentence of the section 2.7 in the original manuscript.

Immunohistochemistry : representative images 3 areas of the should be added

(Response)

For immunohistochemistry, the representative images from the 3 areas (Root side, Middle area, Bone side) have already been provided in the figure as well as in the supplemental figures.

What is the rationale for using an osteoblast cell line (MC3T3-E1): have primary OB cultures been considered? Are the results similar?

(Response)

We appreciate this important comment.

The choice of pre-osteoblast MC3T3-E1 was based on the fact that this cell line has been used in the previous studies assessing osteogenic differentiation. We first tried to use primary rPDL cells for these experiments, but there were large variations in the results, depending on the cells.

The use of primary OB cultures would certainly give us valuable information. We would like to include it in our future study.

what is the hypothesis to explain the fixation of FGF2 on DBBM?

(Response)

It has been reported that adsorption of FGF-2 to the surface of a scaffold occurs via charge interactions and provides a sustained release of bioactive protein from the scaffold (Benington, L.; et al. Pharmaceutics 2020, 12, 508).

In the study of the combination of EMD with DBBM, it was demonstrated that proteins could penetrate inwards through the porous structure of DBBM (Miron, R.J.; et al. J Periodontol. 2012, 83, 936-947. Miron, R.J.; Clin Oral Implants Res. 2017, 28, 327-333.).

Therefore, it was hypothesized that FGF-2 permeates the pores of DBBM and is adsorbed by charge interactions.

ELISA : please add the sensitivity of the kit

(Response)

The assay sensitivity was less than 3 pg/mL. It was added to the section 2.9 of Materials and Methods.

In the results:

the supplementary data 2 are very important and should be included in the main text.

(Response)

We appreciate the advice. We moved the figure from the supplemental files to the main text and present it as Figure 3.

How is the quality of the attachment (anchorage of fibers)? The use of polarized light should be considered (for example: Lallam-Laroye C et al, J Biomed Mater Res, 2006)

(Response)

In the supplemental figure, the formation of a thin layer of cementum-like structure was frequently observed, and the PDL-like collagen bundles were oriented oblique to the root surface in the FGF-2 and FGF-2+DBBM groups at 4 weeks postoperatively. We speculate that FGF-2 enhanced the formation of new connective tissue attachment.

As for the use of polarized light, the observation of collagen using polarized light was performed on undecalcified sections, in the study you mentioned. We used decalcified sections in this study. We would like to prepare undecalcified sections and observe using polarized light in our future study.

the quantification of the histological results have to be done (and not only illustrated using pictures).

(Response)

As for the quantification of the bone formation, we mainly provide results from the micro-CT.

In in vivo experiments assessing periodontal healing or regeneration, creating a reference mark or notch is indeed a standard procedure for the quantification analysis. It is, however, technically difficult to create an appropriate mark or notch on the root of a rat, because of its very small size. Therefore, we did not attempt to quantify the histological results.

For Figure 2, staining for quantification of mineralized bone (Von Kossa?) and osteoid matrix will be very interesting.

(Response)

Thank you for the advice. We felt it is very important to show the general histological results by H-E stained sections. We will consider those suggested staining in our future study.

in 3.7: what are the results on M3C3T1?

(Response)

We only present data for rPDLCs because we are interested in the early events during the healing. In our preliminary experiments, a similar tendency was observed on MC3T3-E1 cells.

in 3.8: on what cells were these results obtained?  (results on both cell types are interesting)

(Response)

We apologize for this. We added cell type information in the Results and the figure legend.